# Alpha-5 Integrin Mediates Simvastatin-Induced Osteogenesis of Bone Marrow Mesenchymal Stem Cells

**DOI:** 10.3390/ijms20030506

**Published:** 2019-01-24

**Authors:** Pei-Lin Shao, Shun-Cheng Wu, Zih-Yin Lin, Mei-Ling Ho, Chung-Hwan Chen, Chau-Zen Wang

**Affiliations:** 1Department of Nursing, Asia University, Taichung 413, Taiwan; m8951016@gmail.com; 2Department of Medical Research, China Medical University Hospital, China Medical University, Taichung 404, Taiwan; 3Orthopaedic Research Center, College of Medicine, Kaohsiung Medical University, Kaohsiung 807, Taiwan; shunchengwu@hotmail.com (S.-C.W.); evolor@gmail.com (Z.-Y.L.); homelin@kmu.edu.tw (M.-L.H.); hwan@kmu.edu.tw (C.-H.C.); 4Department of Orthopedics, Kaohsiung Medical University Hospital, Kaohsiung Medical University, Kaohsiung 807, Taiwan; 5Department of Physiology, College of Medicine, Kaohsiung Medical University, Kaohsiung 807, Taiwan; 6Graduate Institute of Medicine, College of Medicine, Kaohsiung Medical University, Kaohsiung 807, Taiwan; 7Department of Marine Biotechnology and Resources, National Sun Yat-Sen University, Kaohsiung 804, Taiwan; 8Department of Medical Research, Kaohsiung Medical University Hospital, Kaohsiung 807, Taiwan; 9Department of Orthopedics, College of Medicine, Kaohsiung Medical University, Kaohsiung 807, Taiwan; 10Department of Orthopedics, Kaohsiung Municipal Ta-Tung Hospital, Kaohsiung Medical University, Kaohsiung 801, Taiwan; 11Division of Adult Reconstruction Surgery, Department of Orthopedics, Kaohsiung Medical University Hospital, Kaohsiung Medical University, Kaohsiung 807, Taiwan

**Keywords:** bone regeneration, bone marrow mesenchymal stem cells (BMSCs), simvastatin (SVS), osteogenic differentiation, α5 integrin

## Abstract

Simvastatin (SVS) promotes the osteogenic differentiation of mesenchymal stem cells (MSCs) and has been studied for MSC-based bone regeneration. However, the mechanism underlying SVS-induced osteogenesis is not well understood. We hypothesize that α5 integrin mediates SVS-induced osteogenic differentiation. Bone marrow MSCs (BMSCs) derived from BALB/C mice, referred to as D1 cells, were used. Alizarin red S (calcium deposition) and alkaline phosphatase (ALP) staining were used to evaluate SVS-induced osteogenesis of D1 cells. The mRNA expression levels of α5 integrin and osteogenic marker genes (bone morphogenetic protein-2 (BMP-2), runt-related transcription factor 2 (Runx2), collagen type I, ALP and osteocalcin (OC)) were detected using quantitative real-time PCR. Surface-expressed α5 integrin was detected using flow cytometry analysis. Protein expression levels of α5 integrin and phosphorylated focal adhesion kinase (p-FAK), which is downstream of α5 integrin, were detected using Western blotting. siRNA was used to deplete the expression of α5 integrin in D1 cells. The results showed that SVS dose-dependently enhanced the gene expression levels of osteogenic marker genes as well as subsequent ALP activity and calcium deposition in D1 cells. Upregulated p-FAK was accompanied by an increased protein expression level of α5 integrin after SVS treatment. Surface-expressed α5 integrin was also upregulated after SVS treatment. Depletion of α5 integrin expression significantly suppressed SVS-induced osteogenic gene expression levels, ALP activity, and calcium deposition in D1 cells. These results identify a critical role of α5 integrin in SVS-induced osteogenic differentiation of BMSCs, which may suggest a therapeutic strategy to modulate α5 integrin/FAK signaling to promote MSC-based bone regeneration.

## 1. Introduction

Bone injuries are the most common traumatic injuries in humans, and mesenchymal stem cells (MSCs) are considered a valuable source for bone regeneration due to their osteogenic differentiation property [1,2,3,4]. One emerging strategy for bone regeneration is using MSCs combined with active molecules (e.g., MSCs’ secretome, deproteinated bone grafts) to induce osteogenic differentiation to improve bone formation [5,6,7]. However, the capacity of autologous MSCs to differentiate to functional bone-forming osteoblasts remains relatively limited in vivo [3,8]. Therefore, the important issue for MSC-based bone regeneration is finding a new effective strategy to promote the osteogenic differentiation of MSCs.

Statins are lipid-lowering drugs, known as a hydroxy-3-methylglutaryl coenzyme A (HMG-CoA) reductase inhibitor, that are used to inhibit cholesterol biosynthesis [9]. Statins have been reported to enhance the expression of bone morphogenic protein-2 (BMP-2), which is an important growth factor for osteogenic differentiation of stem cells [3,9,10,11,12]. Simvastatin (SVS) is a statin that has been shown to enhance BMP-2 expression and has been clinically studied to stimulate bone formation [9,12,13,14,15,16]. However, the clinical use of SVS for treating bones involves higher clinical doses than those for lipid-lowering therapy, which increases statin-related side effects [12]. Moreover, cell death has been shown when using SVS to induce osteogenic differentiation of MSCs [17]. These findings raise questions regarding the strategy of using SVS to induce osteogenic differentiation of MSCs for bone regeneration.

Clarifying the detailed mechanism of SVS-induced osteogenic differentiation of MSCs may be beneficial for more effective MSC-based bone regeneration. Cell-extra cellular matrix (ECM) interactions, such as those between osteoblasts and the extracellular matrix environment, have been shown to alter osteogenic gene expression and osteogenic differentiation [6,18,19,20]. Cell–ECM interactions involve integrins, a family of transmembrane proteins that induce intracellular signals, which have been shown to contribute to cell spreading, proliferation, differentiation, migration, and survival in different cell types [21,22,23,24,25]. Although several integrins are expressed by osteoblasts and MSCs, information regarding the role of integrins in osteogenic differentiation is limited [5]. Furthermore, whether integrins contribute to SVS-induced osteogenic differentiation of MSCs remains rarely investigated.

Therefore, the role of integrins in SVS-induced osteogenic differentiation in MSCs should be investigated. In this study, we hypothesize that integrins are involved in SVS-induced osteogenic differentiation of MSCs. To test our hypothesis, we investigated integrin expression and integrin signaling during SVS-induced osteogenic differentiation of MSCs. We also tested the role of integrins in SVS-induced osteogenic differentiation of MSCs by siRNA knockdown.

## 2. Results

### 2.1. SVS Increased the mRNA Expression of Osteogenic Markers in D1 Cells

To test the effect of SVS treatment on the osteogenesis of D1 cells, we examined the mRNA expression levels of osteogenic marker genes in D1 cells after SVS treatment. D1 cells were treated with SVS at concentrations of 0.25 and 0.5 µM in basal medium for 3 days. D1 cells were collected, and the mRNA expression levels of osteogenic marker genes were analyzed by Q-PCR. The results show that SVS increased osteogenic gene expression levels in D1 cells. The mRNA expression level of BMP-2 was dose-dependently increased by SVS treatment in D1 cells (Figure 1). The expression levels of marker genes (Runx2, alkaline phosphatase (ALP), and osteocalcin (OC)) involved in osteogenesis were also dose-dependently increased by SVS treatment in D1 cells (Figure 1). Moreover, collagen type I was increased by SVS treatment in D1 cells (Figure 1). These results show that SVS enhanced osteogenic differentiation in D1 cells at concentrations of 0.25 and 0.5 µM.

### 2.2. SVS Enhanced ALP Activity and Calcium Deposition in D1 Cells

To further confirm that SVS treatment enhances osteogenic differentiation in D1 cells, the ALP activity and calcium deposition of D1 cells were tested after SVS treatment. D1 cells were treated with SVS in basal medium at concentrations of 0 (Ctrl), 0.1, 0.25, and 0.5 µM for 3 days, and the medium was changed to osteoinduction for another 5 days. The results showed that ALP activity increased after SVS treatment at concentrations of 0.25 and 0.5 µM (Figure 2A). Alizarin red S staining of D1 cells after SVS treatment also showed that SVS increased the calcium deposition of D1 cells. Compared with the calcium deposition in nontreated control D1 cells (OIM; Ctrl), the calcium deposition of D1 cells was significantly and dose-dependently increased after SVS treatment at concentrations of 0.25 and 0.5 μM (Figure 2B). However, SVS at a concentration of 0.1 μM did not increase the calcium deposition of D1 cells compared with that of the Ctrl (Figure 2B). These results further confirm that SVS enhanced osteogenic differentiation in D1 cells at concentrations of 0.25 and 0.5 µM.

### 2.3. SVS Increased the Expression Levels of α5 Integrin on the Cell Surface of D1 Cells

To investigate the expression levels of integrins on the surface of D1 cells after SVS treatment, D1 cells were treated with SVS at concentrations of 0 μM (Ctrl) or 0.5 μM in basal medium for 3 days. D1 cells were trypsinized and subjected to flow cytometry analysis for detecting β1, β3, α2, αV, and α5 cell surface integrins. Flow cytometry analysis showed that D1 cells expressed αV, α5, and β1 integrins but had lower expression levels of β3 and α2 integrins (Figure 3A). The expression level of α5 integrin on the surface of D1 cells increased by SVS treatment compared with that of the nontreated control (Ctrl) (Figure 3A). The quantification results showed that the expression level of α5 integrin on the cell surface significantly increased after SVS treatment (Figure 3B). Moreover, SVS treatment did not change the expression levels of β1, β3, α2, and αV cell surface integrins in D1 cells compared to those in the nontreated control cells (Ctrl) (Figure 3A). These results showed that SVS increased α5 integrin expression levels on the surface of D1 cells.

### 2.4. SVS Treatment Increased the Phosphorylation of Focal Adhesion Kinase (FAK) and Increased the Protein Expression Level of α5 Integrin in D1 Cells

To confirm whether SVS treatment increases the phosphorylation of FAK the protein expression level of α5 integrin in D1 cells, the phosphorylated focal adhesion kinase (p-FAK)/FAK ratio and total protein expression level of α5 integrin in D1 cells after SVS treatment was also tested. The results showed increased phosphorylation of FAK at 12 and 24 h after SVS treatment. The protein expression levels of the p-FAK/FAK ratio in D1 cells were increased at 12 and 24 h after SVS treatment compared to those in the nontreated control cells (Ctrl) (Figure 4A,B). Moreover, the protein expression level of α5 integrin in D1 cells also increased at 12 h after SVS treatment compared to that in the nontreated control cells (Ctrl) (Figure 4C). These results show that SVS treatment increased the phosphorylation of focal adhesion kinase (FAK) and increased the protein expression level of α5 integrin in D1 cells.

### 2.5. Depletion of α5 Integrin in D1 Cells Using α5 Integrin siRNA

To interfere with α5 integrin expression in D1 cells, Lipofectamine was used to transfect α5 integrin-specific siRNA (si-α5 group) or nonspecific control oligonucleotides (mock group) into D1 cells. D1 cells were transfected with Alexa Fluor Red to evaluate the Lipofectamine transfection efficacy. Figure 5A shows that Alexa Fluor Red was efficiently transfected into D1 cells via Lipofectamine. After culturing in basal medium for 3 days, the mRNA expression levels of α5 integrin in both the mock and si-α5 groups were also tested by real-time PCR analysis. The results showed that the α5 integrin gene expression level in the si-α5 group was significantly decreased in comparison to that in the mock group (Figure 5B).

### 2.6. α5 Integrin Silencing Reduced the mRNA Expression Levels of Osteogenic Marker Genes in D1 Cells during SVS-Induced Osteogenic Differentiation

To investigate whether α5 integrin is involved in SVS-induced osteogenic differentiation, D1 cells in the mock and si-α5 groups were treated with SVS at concentrations of 0 (Ctrl), 0.25 or 0.5 μM in basal medium for 3 days, and osteogenic marker gene expression was evaluated. The Q-PCR results showed that the α5 integrin mRNA expression level was significantly decreased in the si-α5 group at day 3 compared with that in the mock group (Figure 6A). Moreover, the expression levels of osteogenic genes, including BMP-2, Runx2, OC and ALP, were significantly decreased in the si-α5 group at day 3 compared with those in the mock group (Figure 6A). These results show that α5 integrin silencing decreased SVS-induced osteogenic differentiation of D1 cells.

### 2.7. α5 Integrin Silencing Reduced ALP Activity and Calcium Deposition in D1 Cells during SVS-Induced Osteogenic Differentiation

To further confirm that α5 integrin is involved in SVS-induced osteogenic differentiation, ALP activity and calcium deposition in the mock and si-α5 groups were evaluated. D1 cells in the mock and si-α5 groups were treated with SVS at concentrations of 0 (Ctrl), 0.25 or 0.5 μM in basal medium for 3 days and then cultured in osteoinduction medium for another 8 days. The results showed that ALP activity was significantly decreased in the si-α5 group after being cultured in osteoinduction medium for 2 days compared with that in the mock group (Figure 6B). Moreover, the calcium deposition of D1 cells was also significantly decreased in the si-α5 group after being cultured in osteoinduction medium for 6 and 8 days compared with that in the mock group (Figure 6B). These results further confirm that α5 integrin silencing decreased SVS-induced osteogenic differentiation of D1 cells.

## 3. Discussion

Effectively enhancing MSC osteogenic differentiation to engineer new bone is still an unmet need for MSC-based bone regeneration, and the ongoing challenge is to identify key mediators that could optimally promote this process [5,26]. The application of SVS for enhancing the osteogenic differentiation of MSCs is one strategy for stimulating bone formation. Integrins are expressed by osteoblasts and MSCs [5], and they may be an alternative strategy for promoting bone formation other than using SVS. In this study, we determined that α5 integrin plays an important role in SVS-induced osteogenic differentiation in MSCs.

Bone formation involves the commitment of MSCs to the osteoblastic lineage for osteogenic differentiation, first into preosteoblasts and then into mature osteoblasts that synthesize extracellular bone matrix [27,28]. Osteoblast commitment, differentiation, and function are governed by transcription factors that result in the expression of phenotypic genes that lead to acquisition of the osteoblast phenotype [29,30]. Osteogenic differentiation of MSCs is characterized by the expression of the main osteoblast transcription factor (Runx2) and osteoblast markers (ALP, OC, and collagen type I) and is characterized by calcium deposition in the extracellular matrix (ECM) [28,31,32]. BMP-2 is known to promote Runx2 expression in osteogenic differentiation [30,33,34]. Runx2 is the principal transcriptional regulator of osteoblast differentiation and is also required for the expression of ALP and OC during osteogenic differentiation [30,33,34]. We found that SVS treatment increased the mRNA expression levels of BMP-2 and osteogenic marker genes (Runx2, ALP, OC, and collagen type I) at day 3 in basal medium (Figure 1). SVS treatment also increased ALP activity and calcium deposition of D1 cells at day 5 in osteoinduction medium (Figure 2). These results confirm that osteogenic differentiation of D1 cells by SVS treatment can be induced in vitro.

Integrins are transmembrane molecules composed of α and β chains that assemble as heterodimers and are involved in cell–cell and cell–surface adhesion [5]. Binding of external or internal ligands to integrins induces signaling across the plasma membrane [22]. Ligand-integrin binding induces the phosphorylation of FAK and the subsequent activation of key signaling proteins, including phosphatidylinositol 3-kinase (PI3K), mitogen-activated protein kinase (MAPK) ERK1/2, protein kinase C (PKC), and GTPases of the Rho family [35]. A previous study demonstrated that FAK phosphorylation mediates the BMP-2/Smad pathway in terms of stimulating osteogenic differentiation [36,37]. In MC3T3-E1 osteoblastic cells, FAK phosphorylation activates MAPK ERK1/2 and PI3K, leading to the phosphorylation and activation of the key transcription factor Runx2, which results in enhanced osteoblast differentiation [38]. In this study, we showed that α5 integrin expression levels increased in D1 cells after treatment with SVS, while the expression levels of other integrins, including β1, β3, α2, and αV, remained unchanged (Figure 3). SVS treatment increased the activation of FAK at 12 and 24 h, which was accompanied by an increased protein expression level of α5 integrin at 12 h in D1 cells (Figure 4). Based on these results, we speculate that α5 integrin/FAK signaling is particularly required for SVS-induced osteogenic differentiation of MSCs. In the loss-of-function experiments, the silencing of α5 integrin in D1 cells reduced the SVS-induced mRNA expression levels of BMP-2 and osteogenic genes in D1 cells at day 3 and subsequent ALP activity and calcium deposition (Figure 6). These results suggest that α5 integrin/FAK signaling is one of the major regulators in SVS-induced BMP-2 expression and osteogenic differentiation of MSCs.

The limitation of this study is that we did not test the overexpression of α5 integrin in D1 cells for neo-bone formation in vivo. Although we have shown that α5 integrin in MSCs contributes mainly to SVS-induced osteogenic differentiation, whether overexpression of α5 integrin in D1 cells could sufficiently promote neo-bone formation for bone regeneration in vivo requires further investigation. Furthermore, the detailed molecular mechanism of integrins in SVS-induced osteogenic differentiation of MSCs needs to be further investigated. In this study, SVS treatment increased the activation of FAK at 12 and 24 h, which was accompanied by the increased protein expression level of α5 integrin at 12 h in D1 cells. It has been shown that FAK-ERK signaling is necessary for the osteogenic differentiation of MSCs in vitro [39]. Whether FAK-ERK signaling plays an important role in SVS-induced osteogenic differentiation requires further investigation.

## 4. Materials and Methods

### 4.1. Materials

All chemicals were purchased from Sigma-Aldrich (Sigma-Aldrich, St. Louis, MO, USA) unless otherwise specified.

### 4.2. D1 Cells

D1 cells, a mesenchymal stem cell line cloned from bone marrow cells of Balb/c mice (D1 cells, CRL-12424^TM^, ATCC, Manassas, VA, USA), were used in this study [40]. D1 cells were cultured in basal medium (Dulbecco’s modified Eagle’s medium (Gibco BRL, Thermo Fisher Scientific Waltham, MA, USA)) supplemented with 10% fetal bovine serum (FBS), 100 µg/mL sodium ascorbate, 100 mg/mL nonessential amino acids and 100 U/mL penicillin/streptomycin (Gibco BRL, Thermo Fisher Scientific Waltham, MA, USA) in a humidified atmosphere of 5% CO_2_ at 37 °C. D1 cells used in this study are at passage 8–9. For all experiments, D1 cells were cultivated under this condition within 5 to 6 subcultures, and the medium was changed every 2 days. The doubling time of D1 cells was 14 to 16 h under the experimental conditions.

### 4.3. SVS Preparation and Treatment

Simvastatin (SVS) (Merck Sharp and Dohme Corp., Rahway, NJ, USA) was dissolved in dimethyl sulfoxide (DMSO) to prepare stock solutions. All reagents were diluted with basal medium immediately before the treatments began. The SVS concentrations used in these experiments ranged from 0.1 to 0.5 µM. To reduce the influence of DMSO on the D1 cells, the final concentration of DMSO in each treatment was <0.1%. D1 cells cultured in basal medium without any SVS treatment were used as a control (Ctrl). D1 cells were previously seeded into 12-well plates (4 × 10^4^ cells/cm^2^) and cultured for 24 h in basal medium and then treated with SVS. D1 cells were treated with SVS in basal medium for 3 days, and then, the medium was changed to osteoinduction to promote calcium deposition. The osteoinduction medium used in this study consisted of basal medium supplemented with l-ascorbic acid-2-phosphate (50 μM), β-glycerophosphate disodium (10 mM), and dexamethasone (0.1 μM) [41,42]. The medium was changed every 2 days until the D1 cells were harvested. At the indicated time intervals, D1 cells were collected for further experimental analysis.

### 4.4. β1, β3, α2, αV, and α5 Integrin Detection by Flow Cytometry

Flow cytometry analysis was used to examine the presence of β1, β3, α2, αV, and α5 integrins on the cell surface of D1 cells. At the indicated time intervals, D1 cells were harvested from the culture wells by treating them with 0.25% trypsin/EDTA in phosphate buffered saline (PBS). Following the manufacturer’s instructions, one million D1 cells were suspended in 100 μL of PBS containing 10 µg/mL of either phycoerythrin (PE)- or fluorescein isothiocyanate (FITC)-conjugated antibody in 15 mL tubes and incubated at 4 °C for 20 min in the dark. The same isotypes of FITC-conjugated immunoglobulin G1 (IgG1) (BD Biosciences) and PE-conjugated IgG1 (BD Biosciences) were used as negative controls. After incubation for 20 min at 4 °C, the cells were washed 3 times with PBS and then suspended in 1 ml of PBS for analysis. The cell suspensions were analyzed using flow cytometry (Beckman Coulter flow cytometer equipped with a 488 nm argon laser). WinMDI software was used to analyze the flow cytometry results. D1 cells were stained using FITC-conjugated β1 or PE-conjugated α2, α5, αv or β3 antibodies (BD Biosciences).

### 4.5. RNA Isolation and Quantitative Real-Time Polymerase Chain Reaction (qRT-PCR)

At the indicated time intervals, D1 cells were collected. TRIzol reagent (Invitrogen, Carlsbad, CA, USA) was used to extract total RNA from these cells following the manufacturer’s instructions. The RNA quality was confirmed by determining the 260 nm and 280 nm absorbance ratio using a Thermo Scientific NanoDrop^TM^ 1000 spectrophotometer (Thermo Fisher Scientific, Waltham, MA, USA). According to the manufacturer’s instructions, a 260 nm and 280 nm absorbance ratio ranging from 1.8 to 2.0 is considered to indicate no DNA contamination. Subsequently, 0.5–1 μg of total RNA per 20 μL of reaction volume was reverse transcribed into cDNA using a SuperScript First-Strand Synthesis System (Invitrogen, Carlsbad, CA, USA). Real-time PCR reactions were performed and monitored using TOOLS 2× SYBR qPCR Mix (TOOLS, Taiwan) and a quantitative real-time PCR detection system (Bio-Rad Laboratories Inc., Hercules, CA, USA). The cDNA samples (2 µL samples in a total volume of 25 µL per reaction) were analyzed for the genes of interest. The mRNA levels of mouse runt-related transcription factor 2 (Runx2), mouse bone morphogenetic protein-2 (BMP-2), mouse collagen type I (collagen type I), mouse alkaline phosphatase (ALP), mouse osteocalcin (OC), and mouse β-actin (β-actin) were quantified using the following PCR primer pairs: Runx2 (forward: CCC AGC CAC CTT TAC CTA CA; reverse: TAT GGA GTG CTG CTG GTC TG); BMP-2 (forward: AGC TGC AAG AGA CAC CCT TTG; reverse: AGC ATG CCT TAG GGA TTT TGG A); collagen type I (forward: TCA GAG GCG AAG GCA ACA GTC; reverse: GCA GGC GGG AGG TCT TGG); ALP (forward: AAC CCA GAC AGC ATT CC; reverse: GTC AGT CAG GTT GTT CCG ATT CAA); OC (forward: GAG GGC AAT AAG GTA GTG AAC A; reverse: AAG CCA TAC TGG TCT GAT AGC TCG); and β-actin (forward: CCA ACC GTG AAA AGA TGA CC; reverse: ACC AGA GGC ATA CAG GGA CA). The following cycling conditions were used: incubation at 94 °C for 1 min, followed by 35 cycles of denaturation at 94 °C for 30 s, and annealing and extension at 59 °C for 30 s. After real-time PCR reaction, a dissociation (melting) curve was generated to determine the specificity of the reaction. The relative mRNA expression levels of each target gene were calculated from the threshold cycle (*C*t) value of each PCR product and normalized to β-actin expression using the comparative *C*t method [43]. For each gene of interest, readings of four wells from each experimental group at every indicated time point were collected.

### 4.6. Western Blot Analysis

At each indicated time point, D1 cells were washed twice with ice-cold PBS with 1 mM sodium vanadate and lysed in modified radioimmunoprecipitation assay buffer (RIPA; 150 mM NaCl, 1 mM ethylene glycol tetraacetic acid (EGTA), 50 mM Tris, pH 7.4, 10% glycerol, 1% Triton X-100, 1% sodium deoxycholate, 0.1% sodium dodecyl sulfate (SDS) containing a protease inhibitor cocktail (Complete Protease Inhibitor Cocktail Tablets; Roche Diagnostics Ltd., Taiwan) and 1 mM sodium vanadate. The lysates were cleared by centrifugation at 14,000× rpm for 15 min at 4 °C. The expression levels were analyzed by Western blot using antibodies against α5 integrin, p-FAK (Cell Signaling, Danvers, MA, USA), FAK and β-actin (Santa Cruz Biotechnology, Santa Cruz, CA, USA), and they were monitored by enhanced chemiluminescence analysis (ECL system; GE Healthcare, Piscataway, NJ, USA).

### 4.7. α5 Integrin siRNA Transfection

D1 cells were previously seeded into 12-well plates (2 × 10^4^ cells/cm^2^) before siRNA transfection. D1 cells were incubated in antibiotic-free culture medium for 24 h before siRNA transfection. Lipofectamine RNAi MAX reagent (Invitrogen, Carlsbad, CA, USA) was used to transfect α5 integrin siRNA (5 nM) (Ambion) in D1 cells. The RNAi-negative universal control was used (Invitrogen, Carlsbad, CA, USA). Cells were cultured in serum-depleted Opti-MEM medium (Life Technologies, Eugene, OR, USA) during siRNA transfection. To observe the transfection efficiency, Alexa Fluor Red (Invitrogen, Carlsbad, CA, USA) was used for transfection. D1 cells were observed by using a fluorescence microscope, and images were acquired using Image-Pro Plus software, version 5.0 (Media Cybernetics, Silver Spring, MD, USA). After transfection, the culture medium was changed to basal medium for 24 h, followed by treatment with SVS to measure the mRNA expression of osteogenic marker genes and osteogenic differentiation.

### 4.8. Alkaline Phosphatase (ALP) Activity Staining

An ALP kit (No. 85, Sigma-Aldrich, St. Louis, MO, USA) was used to detect ALP activity according to the manufacturer’s instructions. At each indicated time point, cells were fixed with 10% formalin saline at room temperature for 10 min. After washing once with distilled water, the alkaline dye mixture was added to each culture well and incubated for 15 to 30 min. The staining solution was removed, and the wells were washed with distilled water. The fixed and stained plates were then air-dried at room temperature, and the ALP-positive stained cells were photographed.

### 4.9. Alizarin Red S Staining and Quantification

The deposited calcium in a calcified matrix was stained with Alizarin red S staining [44]. At each indicated time point, cells were fixed with 0.05% (*v*/*v*) glutaraldehyde at room temperature for 10 min and then washed with distilled water. The fixed cells were then incubated with Alizarin red S (1% in distilled water, pH 4.2) for 5 min and then extensively washed with distilled water. The fixed and stained plates were then air-dried at room temperature. Stained calcium mineral deposits were photographed. The amount of calcium mineral deposits was determined by dissolving cell-bound Alizarin red S in 10% acetic acid and then quantifying spectrophotometrically at a wavelength of 415 nm.

### 4.10. Statistical Analysis

The data are expressed as the mean ± standard deviation of the mean (SD) of the combined data of experimental replicates. Statistical significance was evaluated using one-way analysis of variance (ANOVA), and multiple comparisons were performed using Scheffe’s method. *p* < 0.05 was considered significant.

## 5. Conclusions

In conclusion, we have shown that α5 integrin mediates SVS-induced osteogenesis of bone marrow mesenchymal stem cells. Therefore, the modulation of α5 integrin/FAK signaling to enhance the osteogenesis of MSCs may be an alternative strategy for MSC-based bone regeneration.

## Figures and Tables

**Figure 1 ijms-20-00506-f001:**
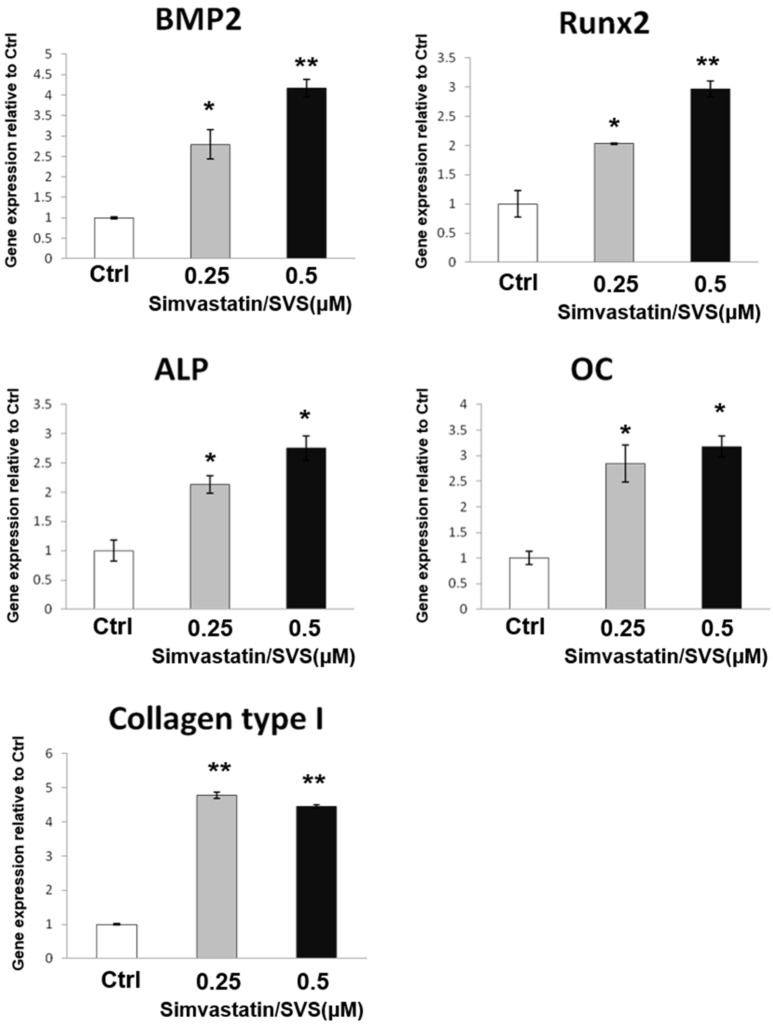
Simvastatin (SVS) increases mRNA expression levels of osteogenic markers in D1 cells. D1 cells (passage 8) were treated with SVS at concentrations of 0 (control: Ctrl), 0.25, and 0.5 µM in basal medium for 3 days. Total RNA was isolated and subjected to real-time PCR analysis. The mRNA expression levels of osteogenic marker genes (bone morphogenetic protein-2 (BMP-2), runt-related transcription factor 2 (Runx2,) alkaline phosphatase (ALP), osteocalcin (OC), and collagen type I) were detected after 3 days of SVS treatment. The gene expression levels are expressed relative to the Ctrl, which is defined as 1. The values presented are the mean ± SD (*n* = 3). * *p* < 0.05 and ** *p* < 0.01 in comparison to the Ctrl.

**Figure 2 ijms-20-00506-f002:**
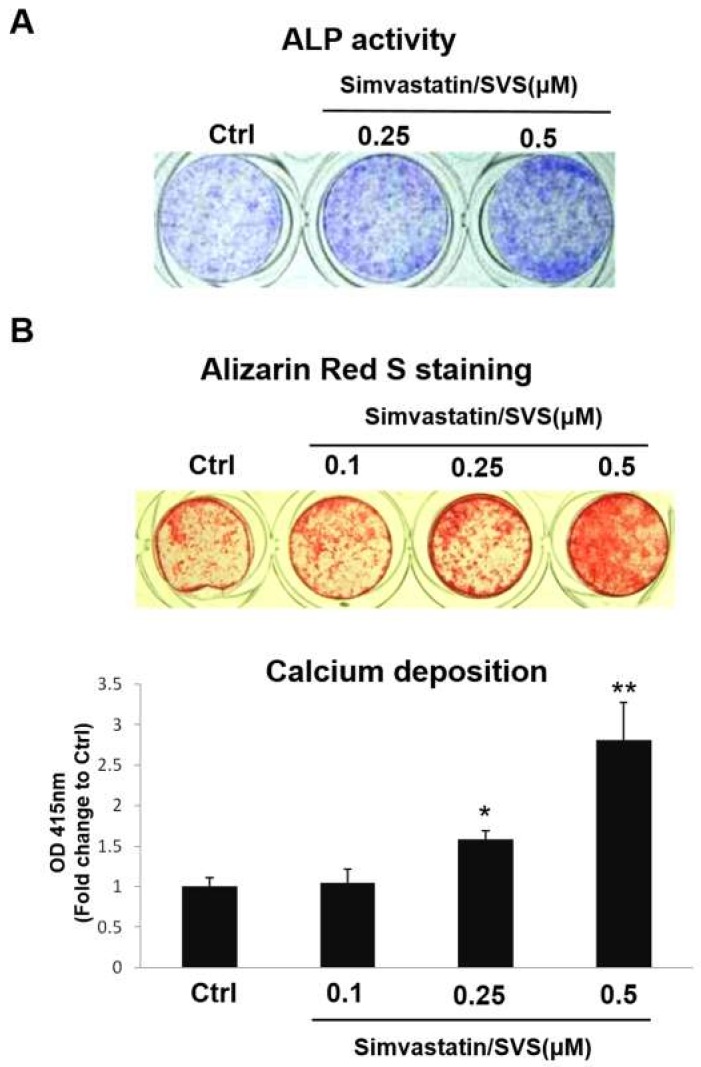
SVS enhances ALP activity and calcium deposition in D1 cells. D1 cells (passage 8) were treated with SVS in basal medium at concentrations of 0 (control: Ctrl), 0.1, 0.25, and 0.5 µM for 3 days, and the culture medium was changed to osteoinduction for an additional 5 days. (**A**) ALP activity staining was detected on day 1 after the medium was replaced by osteoinduction. Blue: staining for ALP activity. (**B**) Alizarin red S staining of calcium deposition was detected on day 5 after the medium was changed to osteoinduction. Red: Alizarin red S staining. The content of calcium deposition is expressed relative to the Ctrl on day 5 after the medium was changed to osteoinduction, which is defined as 1. The values presented are the mean ± SD (*n* = 3). * *p* < 0.05 and ** *p* < 0.01 in comparison to the Ctrl.

**Figure 3 ijms-20-00506-f003:**
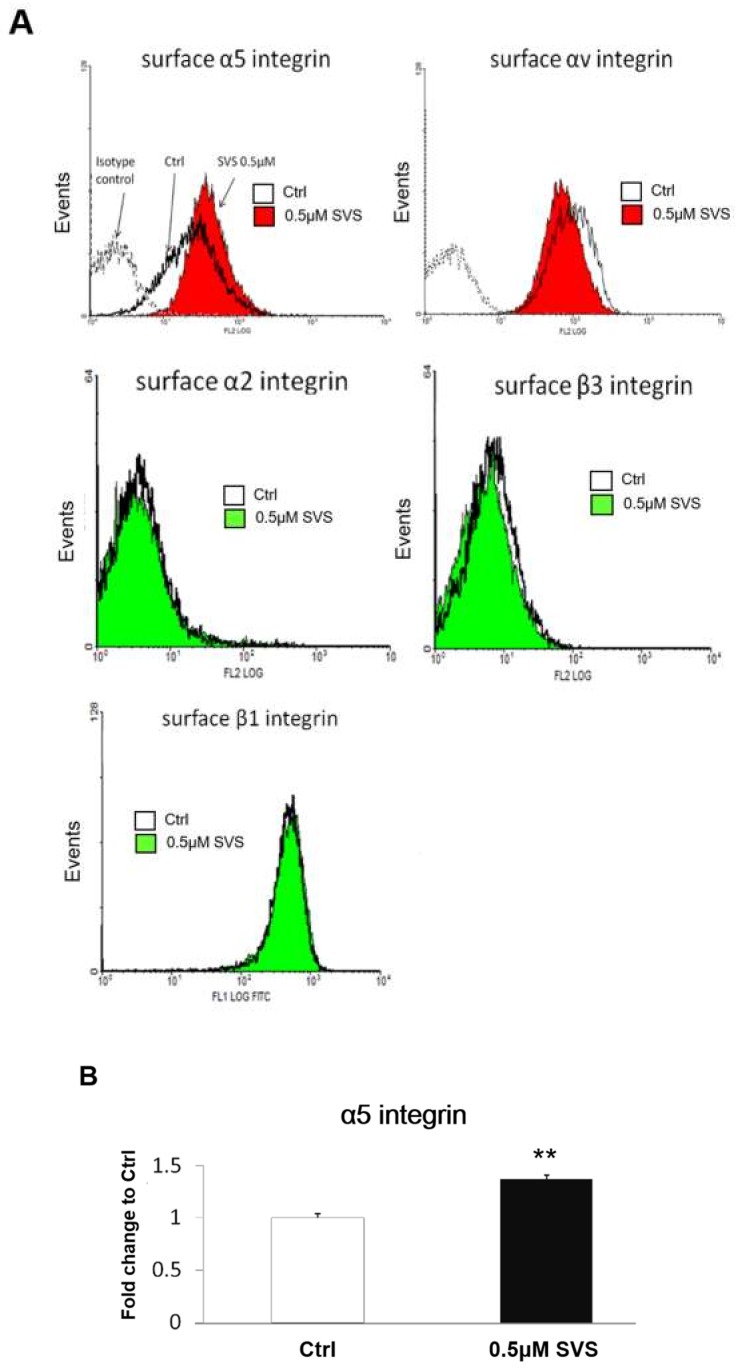
SVS increases the expression of α5 integrin on the surface of D1 cells. D1 cells (passage 8) were treated with SVS in basal medium at concentrations of 0 (control: Ctrl) or 0.5 µM for 3 days. (**A**) Cells were trypsinized and subjected to flow cytometry analysis for detecting α5, αV, α2, β1, and β3 cell surface integrins. The cells were stained with either phycoerythrin (PE)- or fluorescein isothiocyanate (FITC)-conjugated antibodies and analyzed via flow cytometry. FITC-conjugated IgG1 and PE-conjugated IgG1 of the same isotypes were used as isotype controls. (**B**) Quantification of the mean fluorescence of stained surface α5 integrin in D1 cells treated with SVS in basal medium at concentrations of 0 (control: Ctrl) or 0.5 µM SVS for 3 days. The expression level of surface α5 integrin is expressed relative to the Ctrl, which is defined as 1. Data are presented as the mean ± SD (*n* = 3). ** *p* < 0.01 in comparison to the Ctrl.

**Figure 4 ijms-20-00506-f004:**
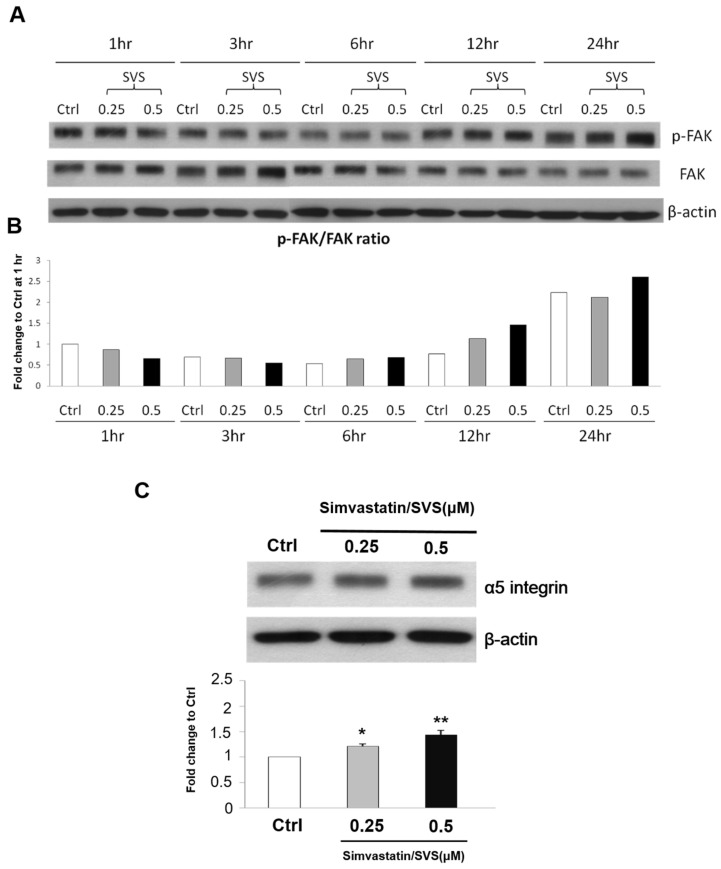
SVS increases the protein expression level of the phosphorylated focal adhesion kinase/ focal adhesion kinase (p-FAK/FAK) ratio and α5 integrin in D1 cells. D1 cells (passage 8) were treated with 0 (Ctrl), 0.25, and 0.5 μM SVS for 1–24 h. (**A**) The protein expression levels of p-FAK and FAK in D1 cells from 1 to 24 h after SVS treatment. (**B**) The ratio of p-FAK and FAK (p-FAK/FAK ratio) is expressed relative to the Ctrl at 1 h after SVS treatment, which is defined as 1. (**C**) The protein expression levels of α5 integrin and β-actin in D1 cells at 12 h after SVS treatment. The protein expression levels of α5 integrin are expressed relative to that of the Ctrl at 12 h, which is defined as 1. The values are presented as the mean ± SD (*n* = 3). * *p* < 0.05 and ** *p* < 0.01 in comparison to the Ctrl group.

**Figure 5 ijms-20-00506-f005:**
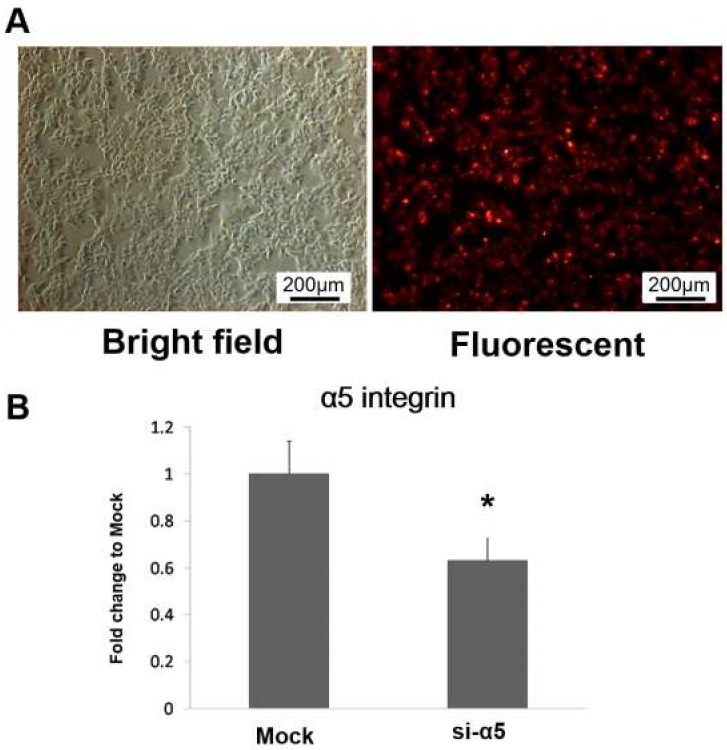
α5 integrin siRNA silencing using Lipofectamine. (**A**) Phase-contrast image and a fluorescent image of D1 cells (passage 9) after Alexa Fluor Red transfection (scale bars = 200 μm). (**B**) α5 integrin mRNA expression levels in D1 cells transfected with nonspecific control oligonucleotides (mock group) or α5 integrin-specific siRNA (si-α5 group) after culture in basal medium for 3 days. The α5 integrin mRNA expression level in the si-α5 group is expressed relative to that in the mock group, which is defined as 1. The values are presented as the mean ± SD (*n* = 3), * *p* < 0.05 in comparison to the mock group.

**Figure 6 ijms-20-00506-f006:**
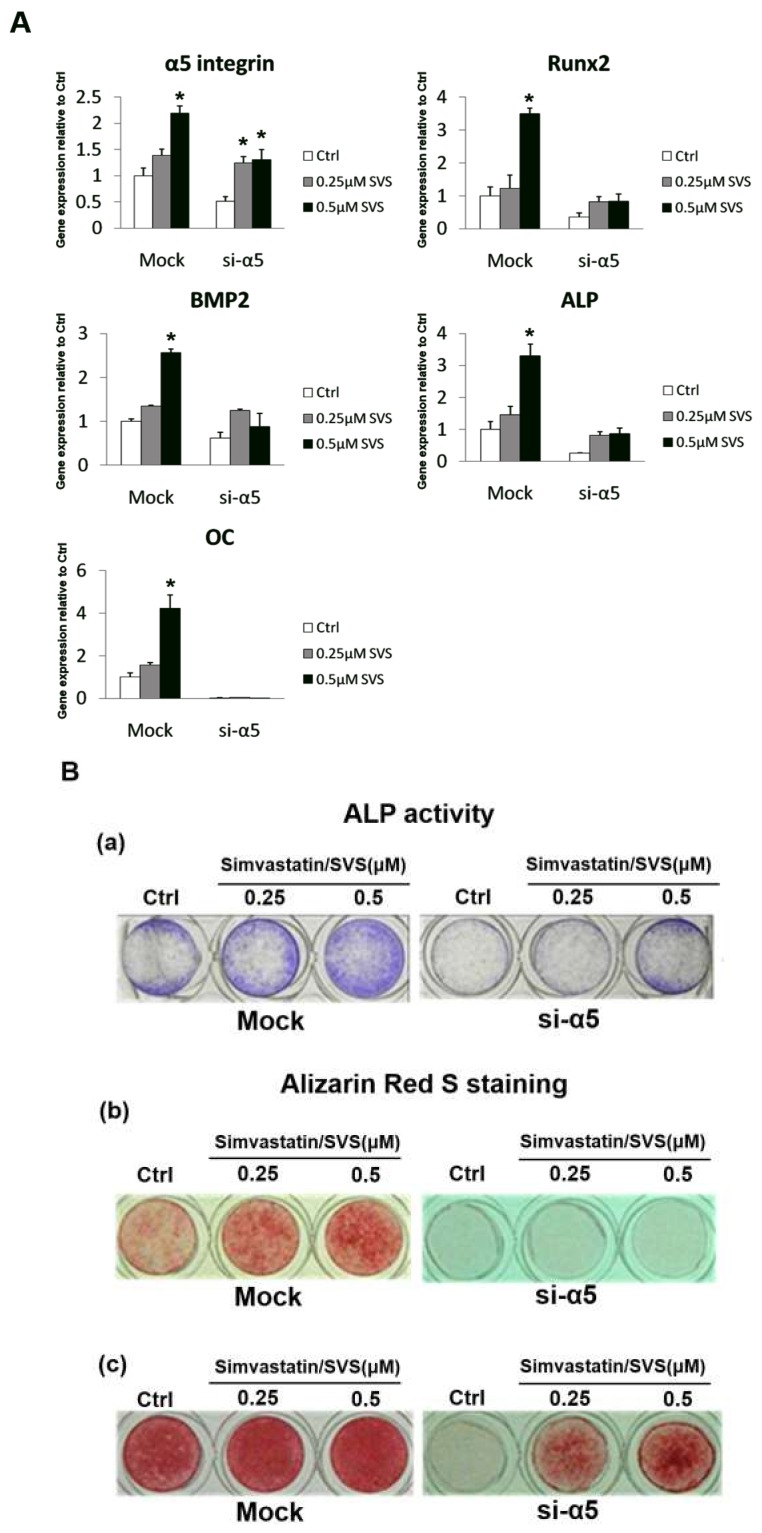
The effect of α5 integrin silencing on SVS-induced mRNA expression levels of osteogenic marker genes, ALP activity and calcium deposition in D1 cells. D1 cells (passage 9) transfected with nonspecific control oligonucleotides (mock group) or α5 integrin-specific siRNA (si-α5 group) were treated with SVS at concentrations of 0 (Ctrl), 0.25 and 0.5 µM for 3 days, and the medium was changed to osteoinduction for another 2 to 6 days. (**A**) The mRNA expression levels of the osteogenic marker genes (α5 integrin, Runx2, BMP-2, ALP, and OC) in the mock and si-α5 group groups at day 3. The gene expression levels are expressed relative to that in the mock group at day 1, which is defined as 1. The values are presented as the mean ± SD (*n* = 3), * *p* < 0.05 in comparison to the mock group. (**B**) ALP activity staining in the mock and si-α5 groups on day 2 after the medium was changed to osteoinduction medium (**a**). Blue: staining for ALP activity. Alizarin red S staining of calcium deposition in the mock and si-α5 groups on day 6 (**b**) and 8 (**c**) after the media was changed to osteoinduction medium. Red: Alizarin red S staining.

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
