# Peer review of "Alpha-5 Integrin Mediates Simvastatin-Induced Osteogenesis of Bone Marrow Mesenchymal Stem Cells"

_ijms, 2019, doi:10.3390/ijms20030506_

Reviewer 1 Report

The only concern regarding figure 5. Authors should improve the image quality.

In the introduction/discussion section authors could improve these section discussing about osteogenic differentiation with other references (e.g. doi: 10.3390/ijms19123916; 10.3390/ijms19020329).

Author Response

Replies to reviewers:

Reviewer 1: (C: Comment, R: Reply)

(C #1) The only concern regarding figure 5. Authors should improve the image quality.

(R #1) Thank you for your comment. We have re-edited figure 5. A clearer image is used to replace the original one.

(C #2) In the introduction/discussion section authors could improve these section discussing about osteogenic differentiation with other references (e.g. doi: 10.3390/ijms19123916; 10.3390/ijms19020329).

(R #2) Thank you for your comment. We have revised introduction section and added more references. The sentence “One emerging strategy for bone regeneration …. to improve bone formation [5].” in the introduction section has been revised into “One emerging strategy for bone regeneration is using MSCs combined with active molecules (e.g., MSCs’ secretome, deproteinated bone grafts) to induce osteogenic differentiation to improve bone formation [5-7].” (line 50-51; marked in red)

The references “doi: 10.3390/ijms19123916; 10.3390/ijms19020329” have been added to the introduction (line 51 and 68; marked in red). These two references have been added as reference 6 and 7 (line 408~ 413; marked in red).

Reviewer 2 Report

Article is interesting by topic and well written.

Introduction is clear and just enough to explain what is the topic, with clear and precise aim of the study.

Methodology is well described in details. One thing only in this part of the article that should be added is what passage of cells was used for each experiment and was it the same for all experiments.

However, the presentation of the results and Figures should be improved. 

For example:

Figure 1 and Figure 6A, in all graphs the y-axis should be labeled as "gene expression relative to Ctrl". In Figure 1, "day 3" in the right corner on the figure should be deleted or  should be replaced with detailed explanation "3-day treatment with simvastatin" or something like that to be clear what is referred to. 

In Figure 2, the labels "day 1" and "day 5" should be changed according to the experimental design, should be "3 days treatment with SVS followed by 1 day cultivation in osteoinduction medium", the same principle should be applied for "day 5" in figure 2B. In figure 2B second part (calcium deposition), on y-axis units should be stated (absorbance?) or how it is measured, or if it is normalized to Ctrl then "fold change to Ctrl". 

In Figure 3A, what M1 and M2 on the histograms means, should be stated in legend. Also, on figure 3B, it is not clear how quantification was performed? were the positive cells counted or mean fluorescence?, y-axis should be changed to "fold change to Ctrl". Also, in figure legend for 3B, it is not written how the quantification is performed. Further, "day 3" in the right corner on the figure should be deleted or  should be replaced with detailed explanation "3-day treatment with simvastatin" or something like that to be clear what is referred to. 

The same comments are applied for Figure 4 as for Figure 2 and 3.

In Figure 5, scale bars don't have a number or it cannot be seen. Also, there are no images of the control culture and they should be included and pointed on the differences and label what can be seen on images (label with asterisks). For Figure 5B, the same comments as for Fig. 2, 3 and 4.

Figure 6B, labels "day 2" "day 6" and "day 8" should be more clear what day mean, according to the explanation in figure legend which is well written.

Author Response

Replies to reviewers:

Reviewer 2: (C: Comment, R: Reply)

(C #1) Methodology is well described in details. One thing only in this part of the article that should be added is what passage of cells was used for each experiment and was it the same for all experiments.

(R#1) Thank you for your comment. The D1 cells used for each experiment in this study are Passage 8-9. The new sentence inMaterials and Methods” has been added (line 281-282; marked in red). The passage of D1 cells used each experiment has been added in figure legends (line 94, 114, 137, 161, 179, 198; marked in red).

(C #2) Figure 1 and Figure 6A, in all graphs the y-axis should be labeled as "gene expression relative to Ctrl". In Figure 1, "day 3" in the right corner on the figure should be deleted or should be replaced with detailed explanation "3-day treatment with simvastatin" or something like that to be clear what is referred to.

(R#2) Thank you for your comment. We have re-edited figure 1 and figure 6A. In figure 1, we have revised the y-axis as "Gene expression relative to Ctrl". The “day 3” in the right corner on the figure 1 is deleted. In figure 6A, we have revised the y-axis as “Gene expression relative to Ctrl”. Both Figure 1 and figure 6A have been replaced by newly edited figures.

(C #3) In Figure 2, the labels "day 1" and "day 5" should be changed according to the experimental design, should be "3 days treatment with SVS followed by 1 day cultivation in osteoinduction medium", the same principle should be applied for "day 5" in figure 2B. In figure 2B second part (calcium deposition), on y-axis units should be stated (absorbance?) or how it is measured, or if it is normalized to Ctrl then "fold change to Ctrl".

(R#3) Thank you for your comment. We have re-edited figure 2 A&B. We have removed the labels "day 1" and "day 5" in figure 2A&B. The y-axis unit is also changed from “Normalized by Ctrl” into “OD 415nm (Fold change to Ctrl)”. Figure 2 has been replaced by newly edited figure.

(C #4) In Figure 3A, what M1 and M2 on the histograms means, should be stated in legend. Also, on figure 3B, it is not clear how quantification was performed? were the positive cells counted or mean fluorescence?, y-axis should be changed to "fold change to Ctrl". Also, in figure legend for 3B, it is not written how the quantification is performed. Further, "day 3" in the right corner on the figure should be deleted or should be replaced with detailed explanation "3-day treatment with simvastatin" or something like that to be clear what is referred to.

(R #4) Thank you for your comment. We have re-edited figure 3 A&B. In Figure 3A, we have removed the M1 and M2 on each histogram, and the “day 3” in the right corner on the figure 3A is also deleted.

In Figure 3B, the quantification of this figure is the mean fluorescence of stained surface α5 integrin in Control and 0.5 µM SVS treated groups in Figure 3A. The y-axis in Figure 3B has been changed as " Fold change to Ctrl ". The “day 3” in the right corner on the figure 3B is also deleted. Figure 3A&B have been replaced by newly edited figure. The figure legend in figure 3 is also revised (line 142~143; marked in red).

(C #5) The same comments are applied for Figure 4 as for Figure 2 and 3.

(R #5) Thank you for your comment. We have re-edited figure 4 B&C. The y-axis in Figure 4B has been changed as " Fold change to Ctrl at 1hr". The y-axis in Figure 4C has been changed as " Fold change to Ctrl ". The “12hr” in the right corner on the figure 4C is deleted. Figure 4 has been replaced by newly edited figure.

(C #6) In Figure 5, scale bars don't have a number or it cannot be seen. Also, there are no images of the control culture and they should be included and pointed on the differences and label what can be seen on images (label with asterisks). For Figure 5B, the same comments as for Fig. 2, 3 and 4.

(R #6) Thank you for your comment. We have re-edited figure 5. A clearer image has been added. The scale bar in Figure 5A has been defined, and the phrase “(200 μm)” has been added in figure 5A. In figure 5A, we apologize for we did not take images of the control culture of D1 cells during Alexa Fluor Red transfection experiment. In Alexa Flour Red experiment, we just want to confirm that siRNA can be transfected into D1 cells, and then we test theα5 integrin mRNA expression levels in D1 cells by qRT-PCR. We showed that α5 integrin mRNA expression levels in D1 cells is successfully down-regulated in si-α5 group in comparison to mock group (Figure 5B). We think that this is supportive that we have successfully down-regulated α5 integrin mRNA expression levels in D1 cells. In figure 5B, the y-axis in Figure 5B has been changed as " Fold change to Mock ". Figure 5 has been replaced by newly edited figure.

(C #7) Figure 6B, labels "day 2" "day 6" and "day 8" should be more clear what day mean, according to the explanation in figure legend which is well written.

(R #7) Thank you for your comment. We have re-edited figure 6B to make it more clear. The labels of day 2, day 6 and day 8 were removed. We add labels of (a), (b), and (c) in Figure 6B. Figure 6B has been replaced by newly edited figure. The explanation in figure legend is also revised (line 206~207; marked in red).